# Sparsity Forcing: Reinforcing Token Sparsity of MLLMs

**Feng Chen**[1][†]  **Yefei He**[2][†]  **Lequan Lin**[3]  **Chenhui Gou**[4]  **Jing Liu**[4]

**Bohan Zhuang**[2][‡]  **Qi Wu**[1]

[1] AIML, University of Adelaide, Australia    [2] ZIP Lab, Zhejiang University, China
[3] University of Sydney, Australia    [4] Monash University, Australia

## Abstract

Sparse attention mechanisms aim to reduce computational overhead with minimal accuracy loss by selectively processing salient tokens. Despite their effectiveness, most methods merely exploit a model's inherent sparsity and thus plateau at moderate budgets (about 50% token reduction), with little headroom to push budget lower without hurting accuracy. Other approaches attempt to enforce sparsity through trainable sparse attention or sharpness-inducing regularizers, but these either fix rigid patterns that ignore input and layer dynamics, or optimize proxy objectives without direct control over token budgets. In this paper, we explicitly reinforce token sparsity in well-posed multimodal large language models (MLLMs) through a simple RL-based post-training framework named *Sparsity Forcing*. Our method explores the efficiency-accuracy trade-off by running multiple rollouts with different token budgets, where both efficiency (token reduction ratio) and performance (answer correctness) are formulated as joint rewards. By contrasting rollouts within each group, the more efficient and correct answers are rewarded while less efficient or incorrect ones are penalized, thereby turning token saving into an end-to-end, inference-consistent optimization objective. Across thirteen image and video benchmarks, Sparsity Forcing raises token reduction ratio on Qwen2-VL/Qwen2.5-VL from 20% to 75% with minimal accuracy decline, significantly reducing long-context inference memory by up to $3\times$ while speeding up decoding by up to $3.3\times$.

## 1 Introduction

Multimodal Large Language Models (MLLMs) (Chen et al., 2024c; Li et al., 2024a; Yang et al., 2024a) have achieved impressive results in tasks such as image captioning and visual question answering (Fu et al., 2023; Li et al., 2024b). However, when processing high-resolution images or long videos, the visual encoder often produces an excessive number of visual tokens, severely constraining the generative efficiency of MLLMs (Chen et al., 2024a). Existing sparse attention methods (Chen et al., 2024a; He et al., 2024a; Jiang et al., 2024) exploit the sparsity of attention maps to prune redundant tokens, thereby reducing memory and latency. The core idea is that many tokens receive negligible attention and can be safely discarded, as in FastV (Chen et al., 2024a), which drops half of the visual tokens after the second layer with little accuracy loss. However, further budget reduction—for instance, to 20% or even 10% of the total tokens—remains challenging, since these approaches merely leverage naturally emergent sparsity of MLLMs rather than actively enforcing it, which fundamentally affects the achievable inference-time acceleration.

Early attempts to enhance token sparsity typically rely on either new attention architectures or sharpness-inducing regularizers on attention maps. For instance, MOBA (Lu et al., 2025) and NSA (Yuan et al., 2025) enforce semi-structured sparsity through trainable sparse attention in LLMs. However, such approaches predefine rigid sparsity patterns and ignore the dynamics across inputs, layers, and training stages. They also require training from scratch, making them less practical when

---

[†]These authors contributed equally. [‡]Corresponding author. Email: `bohan.zhuang@gmail.com`

adapted to MLLMs in a post-training setting. Alternatively, works such as (Gong et al., 2024; Peruzzo et al., 2024; Araabi et al., 2024) introduce structured regularization losses that concentrate attention on semantically relevant regions, thereby reducing token redundancy. Yet these methods optimize proxy objectives like attention sharpness, which does not necessarily translate into consistent end-to-end token savings. Crucially, both streams of work are implemented in the SFT regime under teacher forcing, where sparsity is enforced on ground-truth tokens rather than generated outputs. This creates a mismatch with inference and often limits the realized end-to-end efficiency gains.

In this paper, we propose a reinforcement learning-based post-training method, **Sparsity Forcing**, that directly optimizes the performance-efficiency trade-off of MLLMs via the Group Relative Policy Optimization (GRPO) (Shao et al., 2024). For each vision-language query, we optimize a joint reward that increases with final-answer correctness and with token reduction ratio, *turning token saving into an end-to-end objective rather than a proxy*. Specifically, as shown in Figure 1, we treat an MLLM with sparse attention (*e.g.*, Qwen2-VL+ZipVL (Wang et al., 2024; He et al., 2024a)) as the policy model and the original MLLM with standard causal attention as the reference model; this anchoring stabilizes learning and preserves task fidelity, limiting accuracy loss even at high sparsity. To discover the smallest budget that preserves correctness, we execute multiple rollouts under adaptive token budgets to generate answers—a progressive budget sweep that dynamically tests whether low-salience tokens are actually needed for answer correctness. For each rollout, we compute the joint efficiency-performance reward, then contrast rewards within the group to define the advantages: rollouts that are both correct and more efficient tend to receive positive advantages, while less efficient or incorrect rollouts receive lower or negative advantages; then the policy is updated accordingly. Over training, this update progressively concentrates computation on the most informative tokens. Compared with previous SFT-based methods, our RL-based method applies an *inference-aligned* token pruning policy and KV-cache management for deployment, thereby yielding real end-to-end savings on MLLMs with minimal performance decline.

Our contributions can be summarized as follows:

(1) We propose a novel post-training framework named Sparsity Forcing that explicitly promotes token sparsity in MLLMs, aiming to enhance their achievable efficiency during inference.

(2) We cast efficiency-performance as an explicit joint reward rather than a proxy, producing deployment-aligned sparsity while requiring neither architecture changes nor training from scratch.

(3) Experimental results demonstrate that Sparsity Forcing can enhance the sparsity of Qwen2/2.5-VL from 20% to 75% with a minimal performance decline on thirteen benchmarks, significantly reducing long-context inference memory by up to $3\times$ while speeding up decoding by up to $3.3\times$.

## 2 RELATED WORK

**Sparse attention of MLLMs.** Sparse attention is designed to alleviate the computational constraints during deployment, which arise from the extended visual sequence. A line of work achieves it through token selection. ZipVL (He et al., 2024a) proposed a dynamic ratio allocation strategy of important tokens, which is adaptively determined based on the layer-specific distribution of attention scores. VisionZip (Yang et al., 2024b) is a text-agnostic visual token compressor that selects dominant tokens by attention of the visual encoder and merges the rest by semantic similarity. SparseVLM (Zhang et al., 2024b) selects relevant text "raters" via attention, adaptively prunes visual tokens with a rank-based per-layer ratio, and recycles pruned tokens. SeerAttention (Gao et al., 2024) developed a self-distillation training scheme to efficiently train the AttnGate as a binary classification of token selection. Another line assigns sparsity patterns to attention heads. For example, Minference (Jiang et al., 2024) and MMinference (Li et al., 2025) determined the optimal pattern for each attention head and dynamically built sparse indices based on the assigned pattern during inference. However, these methods primarily leverage inherent attention sparsity at inference rather than actively encouraging token sparsity during optimization, which can lead to accuracy degradation under extremely low budgets. In this paper, we propose Sparsity Forcing, a post-training RL-based framework that explicitly encourages token sparsity by formulating token selection and answer correctness as an efficiency-performance reward-driven decision process that aligns with the actual inference pipeline.

**Encouraging token sparsity.** Recently, trainable sparse attention methods exploit the sparsity patterns of attention during training, which can also be deemed as encouraging token sparsity toward

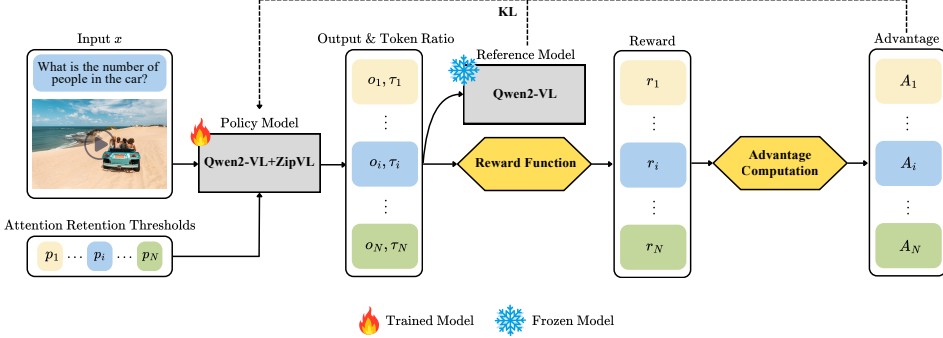

Figure 1: **Overview of the proposed Sparsity Forcing.** We use an MLLM with sparse attention as a policy model, *e.g.*, Qwen2-VL+ZipVL, and the original model with standard causal attention as the reference model. The sampling group is to explore the minimum token ratio required to maintain the current answer under different attention score retention thresholds $p$.

a predefined budget with end-to-end training. For example, NSA (Yuan et al., 2025) designed a dynamic hierarchical sparse strategy, combining coarse-grained token compression with fine-grained token selection to preserve both global context awareness and local precision. MOBA (Lu et al., 2025) proposed a block-wise attention probing method and reformulated the sparse attention as Mixture of Experts. However, these methods typically overlook the dynamics across inputs, layers, and training stages. They also require training from scratch, which makes them less practical and often suboptimal when directly adapted to well-posed MLLMs in a post-training setting. Another line of work explicitly sharpens attention by adding entropy-minimizing or norm-based regularizers to the attention map, *e.g.*, minimum-entropy/$L_\infty$ penalties for NMT (Zhang et al., 2018) and spatial-entropy regularization for ViTs (Peruzzo et al., 2024), which yield sharper, more selective attention distributions. Nevertheless, such sharpening losses optimize a proxy objective: they expose no explicit control over the retained-token budget, and the induced sharpness does not reliably translate into end-to-end token savings or stable accuracy when post-training MLLMs. In this paper, we reformulate the objective as a joint efficiency-performance reward with GRPO and train with multi-budget exploration.

## 3 OUR METHOD

### 3.1 PRELIMINARIES

**Token-level sparse attention.** Let $\mathbf{Q}, \mathbf{K}, \mathbf{V} \in \mathbb{R}^{\ell \times d}$ be query, key, and value matrices in the attention mechanism, respectively, where $\ell$ is the sequence length and $d$ denotes the embedding dimension. In token-level sparse attention, the attention computation is restricted exclusively to a subset of selected tokens in the query and key matrices. Formally, two binary indicator vectors $\mathbf{m}_Q, \mathbf{m}_K \in \{0, 1\}^\ell$ are introduced to specify the selection, where the value of 1 indicates a selected token, and 0 otherwise. For the query and key matrices, the corresponding masks $\mathbf{M}_Q, \mathbf{M}_K \in \mathbb{R}^{\ell \times d}$ are constructed as:

$$\mathbf{M}_Q = \mathbf{m}_Q \cdot \mathbf{1}_d^\top, \quad \mathbf{M}_K = \mathbf{m}_K \cdot \mathbf{1}_d^\top, \tag{1}$$

where $\mathbf{1}_d \in \mathbb{R}^d$ is a vector of ones. The masks then have rows of ones for selected tokens, and zeros elsewhere. Finally, the output of the sparse attention mechanism is computed by:

$$\hat{\mathbf{O}}_{\text{sparse}} = \sigma \left( \frac{(\mathbf{Q} \odot \mathbf{M}_Q)(\mathbf{K} \odot \mathbf{M}_K)^\top}{\sqrt{d}} \right) \mathbf{V}, \tag{2}$$

where $\sigma$ is the row-wise `Softmax` function, and $\odot$ denotes the element-wise matrix multiplication.

The token budget of sparse attention is given by $b = (\|\mathbf{M}_Q\|_0 + \|\mathbf{M}_K\|_0)/d$, where $\|\cdot\|_0$ is the zero-norm function that counts the number of nonzero elements. In this paper, we adopt top-$p$ sparse attention, a mechanism derived from top-$p$ (nucleus) sampling, where the smallest set of tokens whose cumulative attention scores exceeds a threshold $p \in [0, 1]$ is retained for subsequent computation. This design enables dynamic adjustment of sparsity across different layers or tasks while providing a

theoretical upper bound on the approximation error of $(1-p) \times |\mathbf{V}|$ (Lin et al., 2025). Consequently, it supports an online efficiency policy during training with controllable resulting error. Mathematically, the query and key masks are tailored to solve the following constrained optimization problem:

$$\mathbf{M}_Q^*, \mathbf{M}_K^* = \underset{\mathbf{M}_Q, \mathbf{M}_K}{\operatorname{argmin}} \ b, \quad \text{s.t.} \sum_{i=1}^{\ell} \sum_{j=1}^{\ell} \left[ \sigma \left( \frac{\mathbf{Q}\mathbf{K}^\top}{\sqrt{d}} \right) \right]_{ij} (\mathbf{m}_Q)_i (\mathbf{m}_K)_j \geq \ell \times p, \quad (3)$$

where the sum of attention scores equals to $\ell$ due to the row-wise `Softmax` function.

**Inference pipeline with sparse attention.** Given a vision-language question $\mathbf{x} \sim \mathcal{X}$, where $\mathcal{X}$ is the joint input space of multimodal data, MLLMs first prefill the input with sparse attention and save the pruned KV cache $\{\mathbf{K} \odot \mathbf{M}_K, \mathbf{V} \odot \mathbf{M}_K\}$ into memory for decoding. Ideally, the final answer $\mathbf{o}$ is expected to be the same as that without using sparse attention. Since the answer is usually short, that is, $|\mathbf{o}| \ll |\mathbf{x}|$, the final KV cache usage is approximately $2||\mathbf{M}_K||_0/d$. The token ratio is defined as the average proportion of selected tokens per sample, computed as $\tau = \frac{1}{J} \sum_{j=1}^{J} \frac{b_j}{\ell}$ for a sample with token length $\ell$ over all transformer layers $j = 1, ..., J$.

## 3.2 SPARSITY FORCING

**Overview.** As shown in Figure 1, we use an MLLM (*e.g.*, Qwen2-VL (Wang et al., 2024)) with sparse attention (ZipVL (He et al., 2024a)) as our policy model $\pi_\theta$, and the same model with frozen parameters and standard causal attention as the reference model $\pi_{\text{ref}}$. For each question $\mathbf{x} \sim \mathcal{X}$, we perform $N$ independent rollouts through the policy model to generate a set of distinct responses $\{\mathbf{o}_1, ..., \mathbf{o}_N\}$, where each response $\mathbf{o}_n$ is generated with a random threshold $p_n \sim \mathcal{U}(0, 1)$, for $n = 1, ..., N$. In addition, we record the token ratios $\{\tau_1, ..., \tau_N\}$ associated with these responses. Then we compute the reward $r_n$ of each answer with a specially designed reward function that simultaneously considers the performance-efficiency trade-off. As illustrated in Figure 2, sweeping $p$ constitutes a progressive test of whether low-salience tokens are necessary: correctness changes across $p$ reveal the smallest budget that preserves accuracy. Such a more efficient and correct rollout receive a positive advantage in GRPO, while all other rollouts—either incorrect or less efficient—receive less or even a negative advantage. Finally, we use the corresponding advantages to update the policy model. Here are three key benefits of Sparsity Forcing: (1) We make the efficiency-performance trade-off an end-to-end objective rather than a proxy—GRPO jointly rewards token reduction and answer accuracy. (2) Multi-budget rollouts obtained by sweeping thresholds $p$ enable dynamic exploration of the minimum tokens needed for correctness across layers, inputs, and training dynamics, avoiding rigid, hand-crafted positive/negative labels for RL. (3) The training loop mirrors inference: the same token pruning policy and KV-cache management are applied during training and deployment, yielding predictable, budget-aware behavior across tasks and hardware.

**Dynamic sparse attention.** Our approach can be integrated with any sparse attention mechanism. In this paper, we build up our method on top of ZipVL, which is a typical top-$p$ sparse attention. Consider an attention layer with $l$ input tokens, where the standard attention score matrix is denoted as $\mathbf{A} \in \mathbb{R}^{l \times l}$. The accumulated attention score for each token $j$ is calculated by summing the corresponding column:

$$a_j = \sum_{c=1}^{l} \mathbf{A}_{c,j}. \quad (4)$$

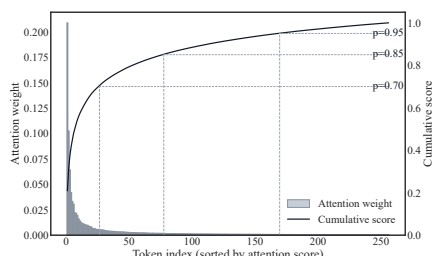

Figure 2: Progressive top-$p$ sampling as a low-salience token test. As $p$ increases, additional tail tokens are included; correctness is then evaluated at each $p$ to identify the minimal budget that preserves accuracy.

These accumulated attention scores are subsequently sorted in descending order, such that $a_{\text{sorted}(j)}$ represents the $j$-th highest attention score. Then, the number of important tokens $b$ is determined by preserving the majority of attention scores with minimal number of tokens, which can be expressed as:

$$b = \min\{p \in \mathbb{Z} \mid \sum_{j=1}^{p} a_{\text{sorted}(j)} \geq p \times l\}. \quad (5)$$

After determining the number of important tokens $b$ for each layer, we use normalized attention scores to assess token importance, calculated as follows:

$$\tilde{a}_j = \frac{\sum_{c=1}^{l} \mathbf{A}_{c,j}}{\text{nnz}(\mathbf{A}_{:,j})}. \tag{6}$$

Here, $\text{nnz}(\mathbf{A}_{:,j})$ denotes the number of non-zero elements in the $j$-th column. Important tokens $\mathbf{T}$ are then selected using the top-$k$ indexing method, while the remainder are considered less important:

$$\mathbf{T} = \text{topk\_index}(\tilde{a}_j, b). \tag{7}$$

Inference is then carried out using only the important tokens in $\mathbf{T}$ for each layer, which participate in subsequent self-attention during both prefilling and decoding, while unimportant tokens are skipped. For implementation, we keep a full KV cache and perform dynamic sparse token selection only at decoding time by fetching the corresponding KV entries on the fly (Kwon et al., 2023).

**Reward function.** The reward of our method is composed of performance and efficiency rewards. Specifically, the performance reward function $r_{\text{per}}$ assigns a binary reward where a correct answer is assigned to 1 and otherwise 0 (Guo et al., 2025). The efficiency reward $r_{\text{eff}}$ is given by the token reduction ratio $1 - \tau_i$, where a higher value indicates more extensive pruning of redundant tokens, thus leading to better efficiency. However, a naive additive combination of them becomes efficiency-biased when no rollout in the group is correct, because the accuracy signal collapses to a constant while the efficiency term still pushes budgets downward. To probe the efficiency-accuracy landscape without collapsing to trivial ultra-sparse policies, we promote efficiency only when the group contains at least one correct answer. Concretely, given a sampling group $\{\mathbf{o}_1, ..., \mathbf{o}_N\}$, we define a group-level indicator:

$$C \triangleq \mathbb{1}\{\exists j \in \{1, \ldots, N\} : \text{Correct}(\mathbf{o}_j) = 1\}. \tag{8}$$

The per-sample reward is then defined as:

$$r_i = r_{\text{per},i} + C \cdot r_{\text{eff},i}, \tag{9}$$

Let $\{r_1, ..., r_N\}$ be the reward of the $N$ generated answers, then the advantage of each answer is obtained by normalizing its reward as:

$$A_i = \frac{r_i - \text{mean}(r_1, r_2, \ldots, r_N)}{\text{std}(r_1, r_2, \ldots, r_N)}. \tag{10}$$

Eventually, the policy update follows the clipped surrogate objective of GRPO (Shao et al., 2024):

$$\mathcal{J}(\theta) = \mathbb{E}_{\substack{\mathbf{x} \sim \mathcal{X} \\ n \in \mathcal{U}([N])}} \left[ \left( \min \left( \frac{\pi_\theta(\mathbf{o}_n \mid \mathbf{x})}{\pi_{\theta_{\text{old}}}(\mathbf{o}_n \mid \mathbf{x})} A_i, \kappa \left( \frac{\pi_\theta(\mathbf{o}_n \mid \mathbf{x})}{\pi_{\theta_{\text{old}}}(\mathbf{o}_n \mid \mathbf{x})} \right) A_i \right) - \beta \mathbb{D}_{\text{KL}}(\pi_\theta \| \pi_{\text{ref}}) \right) \right], \tag{11}$$

where $\theta$ denotes all learnable parameters to be updated; $\mathcal{U}([N])$ is the discrete uniform distribution from 1 to $N$; $\pi_\theta(\cdot)$ and $\pi_{\theta_{\text{old}}}(\cdot)$ are the probabilities under the updated and previous policies, respectively; $\kappa(\cdot)$ is the `clip` operator that constrains the input within a narrow range $(1 - \epsilon, 1 + \epsilon)$ with a small-valued parameter $\epsilon$ to prevent large policy updates; $\mathbb{D}_{\text{KL}}(\pi_\theta \| \pi_{\text{ref}})$ measures the KL divergence between the updated and reference policies; and $\beta$ is the weight coefficient of the KL divergence term. The first term facilitates better responses while ensuring stable updates through clipping. In addition, the second term penalizes deviation from a reference model, hence minimizing the transfer entropy caused by the sparse attention.

## 4 EXPERIMENTAL SETUP

**Implementation.** We adopt QwenVL-series models (Qwen2-VL-7B (Wang et al., 2024), Qwen2.5-VL-3B (Yang et al., 2024a), and Qwen2.5-VL-7B) and LLaVA-series models (LLaVA-Video-7B (Zhang et al., 2024c)) as base MLLMs, owing to their strong adaptation to recent R1-based

Table 1: Performance comparisons with training-free sparse attention on 7 image benchmarks. Here, "Ratio" denotes the average proportion of tokens participating in attention computation over all benchmarks.

| Model | Method | Ratio | MME | MMBench$_{EN}$ | MMStar | ChartQA | TextVQA | OCRBench | MMMU-Pro | Avg. |
|---|---|---|---|---|---|---|---|---|---|---|
| Qwen2-VL-7B | Full | 100 | 2305 | 78.9 | 57.9 | 81.3 | 82.0 | 807 | 32.9 | 70.9 |
| | ZipVL | 79.7 | 2294 | 77.5 | 56.0 | 80.3 | 81.4 | 802 | 32.1 | 69.9 |
| | Sparsity Forcing | 23.6 | 2308 | 78.4 | 58.2 | 81.2 | 82.1 | 807 | 32.9 | 70.8 |
| Qwen2.5-VL-3B | Full | 100 | 2157 | 78.8 | 55.5 | 83.4 | 78.7 | 784 | 31.5 | 69.1 |
| | FastV | 52.1 | 1922 | 75.3 | 52.1 | 78.5 | 76.9 | 737 | 30.2 | 65.1 |
| | VisionZip | 50 | 2026 | 76.6 | 53.5 | 80.5 | 76.4 | 721 | 29.2 | 65.8 |
| | ZipVL | 74.1 | 2142 | 78.1 | 54.3 | 81.4 | 77.6 | 776 | 30.2 | 68.0 |
| | Sparsity Forcing | 22.9 | 2146 | 78.4 | 55.5 | 83.2 | 78.6 | 774 | 30.9 | 68.7 |
| Qwen2.5-VL-7B | Full | 100 | 2303 | 83.9 | 62.2 | 84.0 | 82.9 | 845 | 36.7 | 73.8 |
| | FastV | 52.1 | 2115 | 81.9 | 61.2 | 80.2 | 79.6 | 760 | 34.5 | 69.9 |
| | VisionZip | 50 | 2209 | 83.3 | 61.8 | 80.6 | 79.9 | 796 | 34.6 | 71.2 |
| | ZipVL | 79.5 | 2290 | 83.9 | 60.4 | 82.0 | 82.6 | 837 | 36.2 | 72.9 |
| | Sparsity Forcing | 24.7 | 2286 | 84.1 | 62.5 | 83.1 | 82.6 | 847 | 36.7 | 73.6 |

projects. For QwenVL-series calibration, we employ the Video-R1-260k (Feng et al., 2025) dataset, which covers both image and video modalities to ensure broad domain generalization. For LLaVA-Video, we use a 10k subset of LLaVA-Video-178k (Zhang et al., 2024c) to maintain its capability in the video domain. The parameter $p$ is set within the range of [0.94, 0.975] with a step size of 0.005 for training. We train our model on 8 NVIDIA A100 GPUs with 883 GPU hours for Qwen2.5VL-7b and 164 GPU hours for LLaVA-Video-7b, limiting the number of video frames to 8 for efficiency. The Adam optimizer is used with a learning rate of 1e-6. In the KL divergence term of the GRPO algorithm, the hyperparameter $\beta$ is set to 0.04. During inference, Qwen2.5-VL and Qwen2-VL are evaluated on video benchmarks with approximately 120 frames corresponding to 16,384 tokens, while LLaVA-Video is evaluated with 110 frames corresponding to 20,240 tokens. Notably, during training we employ multiple-budget rollouts to explicitly probe the efficiency-performance trade-off, so that the model learns to enhance token sparsity. At inference, however, we simply fix $p$ to 0.975, which is the upper bound of the optimization range, to guarantee accuracy while still reaping the efficiency gains learned during training.

**Benchmarks.** To assess the effectiveness of our approach, we evaluate on 13 benchmarks: 7 image tasks (MME (Fu et al., 2023), MMBench (Liu et al., 2024a), MMStar (Chen et al., 2024b), ChartQA (Masry et al., 2022), TextVQA (Singh et al., 2019), OCRBench (Liu et al., 2024c), MMMU-Pro (Yue et al., 2024)) and 6 video tasks (VideoMME (Fu et al., 2024), MLVU (Zhou et al., 2024), VideoMMMU (Hu et al., 2025), PerceptionTest (Patraucean et al., 2023), Egoschema (Mangalam et al., 2023), TempCompass (Liu et al., 2024b)). We additionally use HallusionBench (Guan et al., 2024) to assess robustness under low token budgets. Together, these benchmarks span diverse modalities, domains, and difficulty levels for a rigorous evaluation of both efficiency and accuracy. The results are evaluated using LMMs-Eval (Zhang et al., 2024a).

**Baselines of enhancing token sparsity.** We compare our Sparsity Forcing with the existing two baselines to verify its effectiveness in the post-training scenario. *Trainable sparse attention:* We adopt MOBA (Lu et al., 2025) and ZipVL into existing MLLMs to explicitly enforce token sparsity toward a predefined budget/attention retention ratio. The block size of MOBA is set to 256, and the budget is set to 25% for salient token selection. *Attention sharpness loss:* We implement a $L_\infty$ sharpness-inducing regularization term on attention maps (Gong et al., 2024), encouraging the model to focus its attention distribution on a smaller set of highly relevant tokens while suppressing low-importance ones. We adopt the block-wise attention map probing strategy as MOBA to obtain the attention map. During inference, we select 25% salient tokens for sparse attention.

## 5 MAIN RESULTS

**Performance comparison with training-free sparse attention.** As a training-based enhancement of ZipVL, we first compare against ZipVL. As shown in Tables 1 and 2, our Sparsity Forcing can further reduce the token retention ratio of ZipVL from approximately 80% to 25% in QwenVL-series MLLMs and from 45% to 29% on LLaVA-Video across 13 image and video benchmarks with minimal performance decline. Moreover, compared to other training-free sparse attention methods, Sparsity Forcing achieves a much lower token ratio while maintaining accuracy comparable

Table 2: Performance comparisons with training-free sparse attention on 6 video benchmarks. For Minference (Jiang et al., 2024), we report a FLOPs-equivalent token ratio.

| Model | Method | Ratio | VideoMME | MLVU$_{\text{M-avg}}$ | VideoMMMU | PerceptionTest | EgoSchema | TempCompass | Avg. |
|---|---|---|---|---|---|---|---|---|---|
| Qwen2-VL-7B | Full | 100 | 62.9/68.2 | 66.4 | 42.2 | 61.6 | 66.2 | 67.1 | 62.1 |
| | ZipVL | 71.2 | 62.1/67.8 | 64.9 | 42.1 | 61.3 | 66.0 | 66.8 | 61.6 |
| | Sparsity Forcing | 23.8 | 61.8/68.3 | 66.0 | 42.4 | 61.3 | 66.2 | 67.3 | 61.9 |
| Qwen2.5-VL-3B | Full | 100 | 61.2/66.8 | 67.3 | 41.8 | 66.4 | 64.2 | 64.0 | 61.7 |
| | ZipVL | 69.4 | 60.6/66.5 | 67.3 | 41.4 | 65.2 | 63.7 | 63.9 | 61.2 |
| | Sparsity Forcing | 22.8 | 61.0/66.5 | 67.2 | 41.9 | 66.2 | 64.0 | 63.7 | 61.5 |
| Qwen2.5-VL-7B | Full | 100 | 64.5/71.1 | 69.7 | 46.3 | 69.0 | 64.3 | 71.2 | 65.2 |
| | ZipVL | 70.3 | 64.1/70.8 | 69.5 | 45.9 | 68.9 | 64.0 | 70.6 | 64.8 |
| | Sparsity Forcing | 25.9 | 64.0/71.0 | 69.5 | 46.2 | 68.7 | 64.2 | 70.9 | 64.9 |
| LLaVa-Video-7B | Full | 100 | 64.5/70.8 | 70.4 | 35.2 | 66.1 | 57.0 | 66.1 | 61.4 |
| | FastV | 51.6 | 64.0/70.5 | 69.7 | 34.8 | 65.8 | 56.2 | 65.4 | 60.9 |
| | VisionZip | 50 | 42.2/60.6 | 52.9 | 31.7 | 42.0 | 39.7 | 46.2 | 45.0 |
| | Minference | 46.1 | 64.3/70.6 | 70.5 | 34.8 | 65.9 | 56.8 | 66.0 | 61.3 |
| | ZipVL | 45.2 | 64.3/70.3 | 70.2 | 34.8 | 65.2 | 56.7 | 65.7 | 61.0 |
| | Sparsity Forcing | 29.6 | 64.1/70.2 | 70.3 | 35.2 | 66.0 | 57.0 | 66.2 | 61.3 |

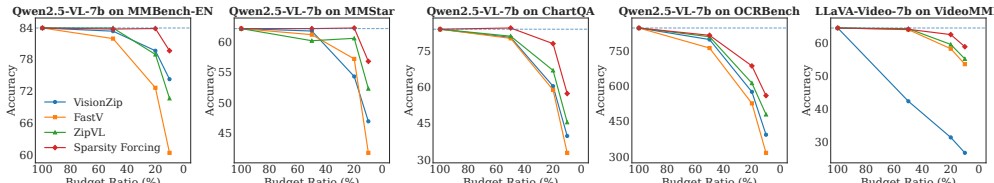

Figure 3: Adjustment under low budgets across different models and benchmarks.

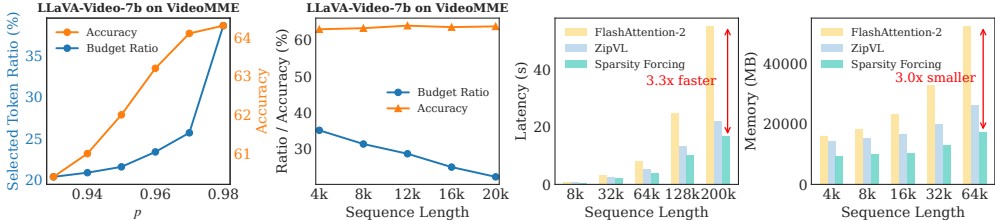

Figure 4: (a) The effect of attention scores retention threshold $p$ on token ratio and performance. (b) Accuracy and token budget with respect to increasing token sequence. (c)(d) Prefill latency and decoding memory usage under varying sequence lengths on LLaVA-Video-7b.

to the standard-attention baseline, thus demonstrating a superior balance between efficiency and performance. On LLaVA-Video-7B, it matches Minference's (Jiang et al., 2024) accuracy while using 29.6% tokens vs. 46.1% for Minference. Moreover, it clearly outperforms VisionZip (Yang et al., 2024b) and FastV (Chen et al., 2024a), which both require around 50% tokens but achieve lower or similar accuracy. These results show that Sparsity Forcing can significantly reduce computation by enhancing token sparsity while maintaining performance across image and video benchmarks.

**Performance comparison with post-training baseline methods in enhancing token sparsity.** As shown in Table 3, Sparsity Forcing achieves a superior efficiency-performance trade-off compared to baseline methods on Qwen2.5-VL-7B. The standard-attention model reaches an average score of 73.2, while MOBA and sharpness loss, both operating at a 25% token ratio, suffer from noticeable drops to 66.6 and 67.8, respectively. ZipVL provides a moderate balance, obtaining 71.5 at a higher token ratio of 61.7%. In contrast, Sparsity Forcing attains 72.8

Table 3: Comparisons with baseline methods of enhancing token sparsity on Qwen2.5VL-7b. † denotes post-training MLLMs with ZipVL.

| Method | Ratio | MME | MMStar | ChartQA | VideoMME | Avg |
|---|---|---|---|---|---|---|
| Full | 100 | 2303 | 62.2 | 84.0 | 64.5 | 73.2 |
| MOBA | 25.0 | 1906 | 58.6 | 77.3 | 62.6 | 66.6 |
| ZipVL† | 61.7 | 2264 | 62.0 | 78.9 | 64.2 | 71.5 |
| Sharpness loss | 25.0 | 1965 | 59.6 | 77.0 | 63.7 | 67.6 |
| Sparsity Forcing | 26.4 | 2286 | 62.5 | 83.1 | 64.0 | 72.8 |

even with only 26.4% tokens, outperforming all baseline approaches on tasks such as MMStar (Chen et al., 2024b) and ChartQA (Masry et al., 2022). These results highlight the effectiveness of our RL-based optimization that reformulates the efficiency-performance tradeoff as explicit rewards.

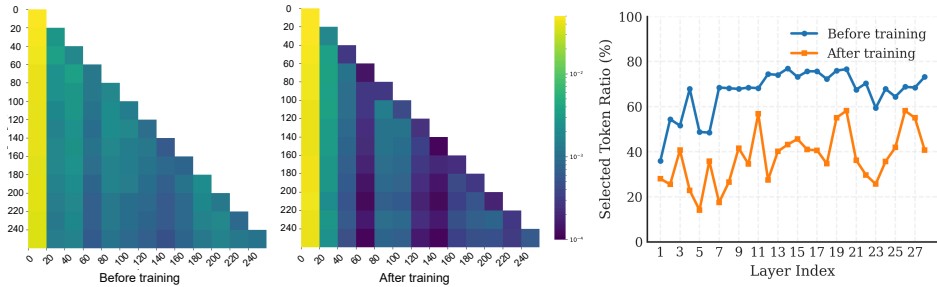

Figure 5: Token sparsity evolution of Qwen2.5-VL-7b. Left/center: attention maps of the same sample before vs. after training, suggesting our method reinforces MLLM to concentrate on a smaller subset of tokens. Right: layer-wise selected token ratio before and after training, showing that the achievable sparsity differs markedly across layers.

Table 4: Ablation study on different sparse attention with top-$k$, top-$p$, and threshold-based pruning.

| Model | Sparse attention | Ratio | MME | VideoMME |
|---|---|---|---|---|
| | Full | 100 | 2303 | 64.5 |
| Qwen2.5-VL-7B | Top-$k$ | 25 | 2160 | 60.2 |
| | Threshold | 37.8 | 2218 | 61.6 |
| | Top-$p$ | 24.1 | 2286 | 64.0 |

Table 5: Ablation study of different ranges of $p$ and group sizes for training.

| Range ($p$) | Group size | Ratio | MME | VideoMME |
|---|---|---|---|---|
| $[0.94, 0.975]$ | 8 | 24.1 | 2286 | 64.0 |
| $[0.94, 0.975]$ | 12 | 23.7 | 2290 | 63.9 |
| $[0.80, 0.95]$ | 8 | 29.0 | 2285 | 63.6 |
| $[0.80, 0.95]$ | 12 | 25.7 | 2285 | 63.7 |

**Adjustment under different budgets.** We further evaluate Sparsity Forcing under varying token budgets to illustrate its adaptability. As shown in Figure 3, Sparsity Forcing enables flexible budget adjustment across tasks and models. On commonsense benchmarks (Liu et al., 2024a), it maintains strong performance even under an extremely low budget of about 10% tokens. On OCR-related tasks (Liu et al., 2024c; Masry et al., 2022), it can further compress token ratio while still preserving better accuracy than other methods, demonstrating its ability to focus on fine-grained textual regions. Moreover, this property generalizes well across both image and video benchmarks and different model backbones, outperforming other sparse attention baselines under the same token ratio.

Besides, we further discuss the influence of the attention retention ratio $p$ during inference. As shown in Figure 4 (a), increasing $p$ smoothly raises the token ratio while also improving accuracy, even for the value outside the optimization range, showing that Sparsity Forcing enables controllable efficiency-accuracy trade-offs without abrupt performance drops.

**Runtime speed comparison.** As shown in Figure 4 (c)(d), Sparsity Forcing delivers clear efficiency gains over existing methods. It achieves up to $3.3\times$ faster inference and $3.0\times$ lower memory usage than FlashAttention-2 (Dao, 2023) at 200k sequence lengths. These results highlight that Sparsity Forcing not only reduces token budgets but also substantially improves runtime efficiency for large-scale multimodal inference.

## 6 ABLATION STUDY AND DISCUSSION

**Sparsity Forcing with different sparse attention mechanisms.** Table 4 compares Sparsity Forcing using various sparse attention strategies, including top-$k$ sampling, threshold-based pruning, and dynamic top-$p$ sampling. While top-$k$ and threshold-based methods can reduce token usage, they rely on fixed sparsity patterns or constraints and thus act as *offline policies*, which may not adapt optimally to input-specific variations. In contrast, top-$p$ sampling enables *online* adjustment of token retention according to attention score distributions, achieving the best trade-off between efficiency and performance (*e.g.*, 24.1% token ratio with minimal accuracy drop). This highlights the advantage of coupling grouped-rollout RL with a dynamic, inference-time controllable sparse attention mechanism.

**Token sparsity change.** Figure 5 shows how training reshapes both the attention distribution and the retained token ratio. After training, the model concentrates attention on a more compact subset of tokens, as illustrated in the left and center figures, demonstrating effective pruning. The layer-wise

curves (right) reveal substantial variation in attainable sparsity across depth, indicating that a single global retention rate is suboptimal and a dynamic, layer-aware policy is preferable.

**Robustness using a low token budget.** Aggressive token pruning can drop essential evidence and increase hallucination risk. To assess robustness, we conduct a sensitivity analysis on HallusionBench (Guan et al., 2024) under a strict token budget. As shown in Figure 6, Sparsity Forcing closely tracks the original model across all accuracy, figure accuracy, and question pair accuracy (aAcc, fAcc, and qAcc) with only minor fluctuations, indicating that our dynamic sparse attention preserves key evidence and does not amplify hallucinations in low-budget settings.

**Scalable token sparsity.** We examine how learned token sparsity scales with model capacity and context length. As shown in Tables 1 and 2, we find models of different sizes and capabilities in the same MLLM family exhibit broadly similar retained token ratios, a phenomenon that holds across both image and video domains. Moreover, in Figure 4 (b), as the input sequence grows from 4k to 20k tokens, the retained-token ratio steadily decreases while accuracy remains nearly unchanged. This suggests that learned token sparsity adapts to longer sequences by safely discarding more redundant tokens yet preserving essential information, making it particularly effective for high-complexity, long-context video understanding.

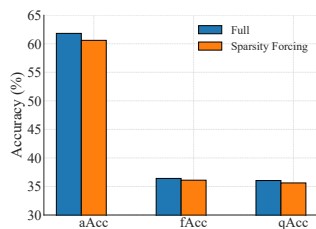

Figure 6: Sensitivity analysis of Qwen2.5-VL-7b on HallusionBench using low token budgets.

**Why grouped-rollout RL?** Our method requires comparing multiple budgets per query to explore the efficiency-performance trade-off, where the notion of "positive vs. negative" is inherently non-stationary: as training progresses, sparsity patterns and the minimal correct budget shift, so predefined preference pairs of DPO (Rafailov et al., 2023) and rigid sparse pattern of trainable sparse attention quickly become misaligned with the current trade-off frontier. In contrast, grouped-rollout RL methods (*e.g.*, GRPO, DAPO (Yu et al., 2025)) operate on-policy and define relative advantages within each sampled group of rollouts. This design naturally provides exploration and credit assignment directly tied to budget choices, and avoids maintaining a large, ever-updating preference dataset.

**Budget sampling strategy.** As shown in Table 5, the ablation of the range of $p$ and group size can be interpreted as probing the efficiency-performance trade-off through grouped rollouts. Increasing the group size enables denser coverage of candidate token budgets, allowing the model to explore the budget–accuracy landscape more thoroughly and providing stronger contrastive signals for learning sparsity. However, this benefit comes at the cost of higher training overhead, since more rollouts must be generated per sample. In addition, setting the $p$ range relatively high improves the likelihood that at least one rollout within the group is both correct and efficient, which is crucial for contrastive learning to effectively guide sparsity.

## 7 CONCLUSION

In this paper, we presented Sparsity Forcing, an RL-based post-training method that explicitly encourages token sparsity in well-posed MLLMs, delivering substantial inference-time acceleration. We treat the token-reduction ratio as the efficiency reward and answer accuracy as the performance reward. Concretely, Sparsity Forcing performs multi-budget rollouts and applies GRPO with group-wise advantages to favor the smallest budget that still yields a correct answer. Compared with previous SFT-based methods, our RL-based approach enforces an inference-aligned token pruning policy and KV-cache management, thereby delivering real end-to-end efficiency gains in MLLMs while keeping performance degradation minimal. Experiments across thirteen image and video benchmarks show that Sparsity Forcing increases the token reduction ratio on Qwen2-VL/Qwen2.5-VL from 20% to as high as 75% with minimal accuracy loss, while cutting long-context inference memory by up to $3\times$ and speeding up decoding by up to $3.3\times$.

**Limitations and future work.** Our current work mainly focuses on single-turn hardware-agnostic token sparsity. In our future work, we plan to extend reinforced efficiency to broader applications and objectives—such as hardware-aware targets (latency/memory/energy), multi-turn dialogue, tool-/retrieval-call budgets, head/layer/MoE expert gating, and KV/activation quantization and retention.

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

# A  APPENDIX

## A.1  ALGORITHM OF OUR DYNAMIC SPARSE ATTENTION

As shown in Algorithms 1 and 2, we summarize the sparse attention mechanisms used in our method for the prefill and decoding.

---

**Algorithm 1:** Prefill phase of our dynamic sparse attention.

---

**Input:** Input embedding $\mathbf{X}$, attention retention ratio $p$.
**Output:** Attention output $\mathbf{O}$, KV cache $(\mathbf{K}, \mathbf{V})$.
Calculate query, key and value states $(\mathbf{Q}, \mathbf{K}, \mathbf{V})$
```
// Probe attention map
```
Select a subset of tokens $\mathbf{Q}'$ from query states and compute attention scores $\mathbf{A}' = \text{Softmax}\left(\mathbf{Q}'\mathbf{K}^T\right)$
Determine the number of important tokens as per Eq. (5)
Calculate the normalized attention scores for each token as per Eq. (6)
Select a set of important tokens $\mathbf{T}$ as per Eq. (7)
```
// Token-level Sparse
   Attention with
   FlashAttention
```
$\mathbf{O} = \text{FlashAttention}(\mathbf{Q}[\mathbf{T}], \mathbf{K}[\mathbf{T}], \mathbf{V}[\mathbf{T}])$
```
// KV Cache
```
$\mathbf{K} = \mathbf{K}[\mathbf{T}]$
$\mathbf{V} = \mathbf{V}[\mathbf{T}]$
**return** $\mathbf{O}$, *($\mathbf{K}$, $\mathbf{V}$)*

---

**Algorithm 2:** Decoding phase of our dynamic sparse attention.

---

**Input:** Input embedding $\mathbf{x}$, stored KV cache $(\mathbf{K}_{\text{in}}, \mathbf{V}_{\text{in}})$.
**Output:** Attention output $\mathbf{o}$, updated KV cache $(\mathbf{K}_{\text{out}}, \mathbf{V}_{\text{out}})$.
Calculate query, key and value states $(\mathbf{q}, \mathbf{k}, \mathbf{v})$
```
// Fetch KV cache from memory
   and update
```
$\mathbf{K}_{\text{out}} = \text{Concat}(\mathbf{K}_{\text{in}}, \mathbf{k}), \quad \mathbf{V}_{\text{out}} = \text{Concat}(\mathbf{V}_{\text{in}}, \mathbf{v})$
```
// Compute attention output
```
$\mathbf{o} = \text{FlashAttention}(\mathbf{q}, \mathbf{K}_{\text{out}}, \mathbf{V}_{\text{out}})$
**return** $\mathbf{o}$, *($\mathbf{K}_{\text{out}}$, $\mathbf{V}_{\text{out}}$)*

---

## A.2  IMPLEMENTATION OF ATTENTION SHARPNESS LOSS

Following prior works such as MOBA (Lu et al., 2025) and NSA (Yuan et al., 2025), we adopt a block-wise probing strategy by applying average pooling over queries and keys with block size $B$ to approximate the attention map. Given the visual input $\mathbf{X}_v$ with query $\mathbf{Q}_v$ and key $\mathbf{K}_v \in \mathbb{R}^{l \times d}$, we first compress the sequence via average pooling:

$$\bar{\mathbf{Q}}_v = \text{mean\_pool}(\mathbf{Q}_v, B), \quad \bar{\mathbf{K}}_v = \text{mean\_pool}(\mathbf{K}_v, B),$$

where $\text{mean\_pool}$ is applied along the sequence dimension, yielding compressed representations $\bar{\mathbf{Q}}_v, \bar{\mathbf{K}}_v \in \mathbb{R}^{l/B \times d_i}$. The block-wise attention map is then computed as

$$\bar{A} = \sigma\left(\frac{\bar{\mathbf{Q}}_v \bar{\mathbf{K}}_v^T}{d_i} + \mathcal{M}\right),$$

where $\sigma$ denotes the $\text{softmax}$ function and $\mathcal{M}$ is the causal attention mask.

To promote sparsity, we encourage each query to concentrate on a few salient keys by introducing an $L_\infty$ sharpness-inducing regularization:

$$\mathcal{L}_{\text{sharp}} = -\frac{1}{l/B} \sum_{i=1}^{l/B} \|\bar{A}_i\|_\infty,$$

where $\bar{A}_i$ denotes the $i$-th row of $\bar{A}$. This loss increases the peak mass of each query's attention distribution, enforcing sharper focus and reducing redundancy among tokens. At inference time, we apply sparse attention by retaining only the top 25% salient tokens according to $\bar{A}$.

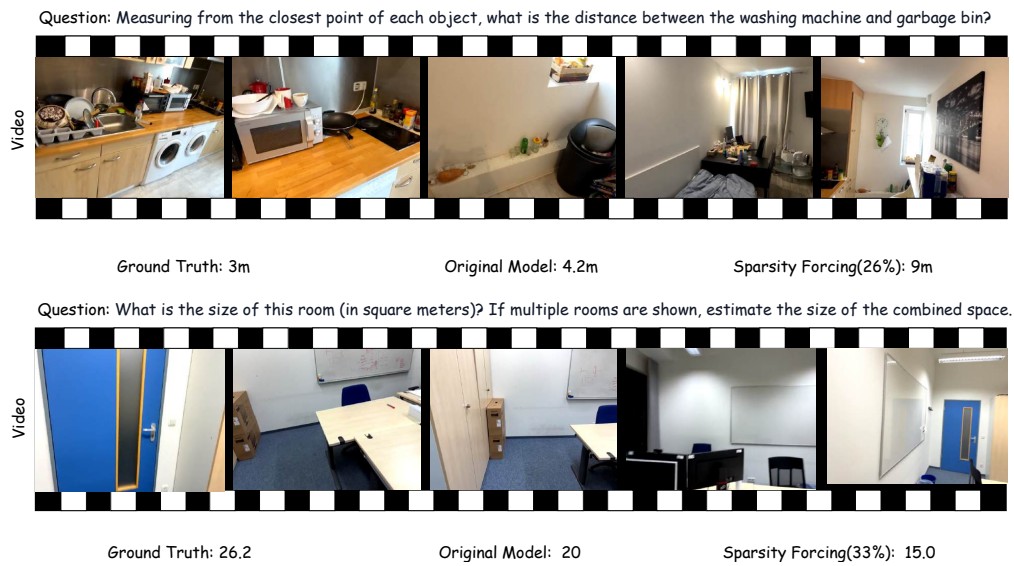

Figure 7: Failure cases. Our method still struggles with spatial reasoning, particularly in estimating relative object distances and available space. Results are shown on Qwen2.5-VL-7B.

### A.3 IMPLEMENTATION OF SPARSITY FORCING WITH DIFFERENT SPARSE ATTENTION

To implement Sparsity Forcing with different sparse attention mechanisms, we first compute the accumulated attention score $a$ of each token as in Eq. 4, and normalize it to $\tilde{a}$ using Eq. 6, since early tokens accumulate higher values than later ones (He et al., 2024b). For top-$k$ sparse attention, we retain the $k$ most salient tokens ranked by $\tilde{a}$ for subsequent attention. For threshold-based sparse attention with threshold $\tau$, we keep all tokens with $\tilde{a} > \tau$.

During training, we adopt multiple budget rollouts to probe the efficiency-accuracy trade-off. For top-$k$ sampling, we set $k$ within [25%, 50%] at intervals of 5%, where 25% reflects the target efficiency budget while 50% corresponds to the accuracy-preserving range of most sparse attention methods. For threshold-based attention, we empirically vary $\tau$ within $[10^{-3}, 10^{-2}]$ with a step of 0.002. At inference time, we fix the budget by retaining the top 25% salient tokens for top-$k$ sparse attention and set $\tau = 10^{-3}$ for threshold-based sparse attention.

### A.4 FAILURE CASE

These failure cases in Figure 7 highlight a key limitation of token pruning, especially with enhanced sparsity: while it effectively reduces redundancy, it may inadvertently disrupt cross-frame correspondences that are critical for spatial reasoning. Unlike commonsense reasoning, which often relies on text-grounded cues, spatial understanding demands alignment of visual features across frames—such as consistent tracking of object positions, relative distances, and scene layouts.

### A.5 MORE RESULTS AND DISCUSSION

Table 6: Ablation of different efficiency reward designs on LLaVA-Video-7B.

| Efficiency reward | Ratio | VideoMME | MLVU$_{M\text{-avg}}$ | VideoMMMU | PerceptionTest | EgoSchema | TempCompass | Avg. |
|---|---|---|---|---|---|---|---|---|
| Token reduction ratio | 29.6 | 64.1/70.2 | 70.3 | 35.2 | 66.0 | 57.0 | 66.2 | 61.3 |
| Latency + token reduction ratio | 30.5 | 63.9/70.3 | 70.4 | 35.0 | 66.4 | 56.4 | 66.0 | 61.2 |

**Hardware efficiency reward.** Hardware latency is an important practical metric. We therefore conducted an additional ablation in which we explicitly include measured hardware latency in the efficiency reward. Concretely, we compare two variants on LLaVA-Video-7B: (i) using only the

token reduction ratio as the efficiency term, and (ii) using a combined efficiency reward that depends on both latency and token reduction ratio. As shown in Table 6, incorporating latency into the reward does not yield noticeable benefits: both variants achieve almost identical accuracy (61.3 vs. 61.2 Avg.) with very similar token ratios. These results suggest that the token reduction ratio is already a sufficiently good surrogate for hardware efficiency in our setting.

Table 7: Training and rollout cost comparison with baseline methods. Latency is measured under a 128k context.

| Method | Wall-clock training time (hour) | Latency(s) |
|---|---|---|
| MOBA | 75.6 | 11.6 |
| ZipVL | 81.0 | 18.7 |
| Sharpness loss | 88.4 | 11.9 |
| Sparsity Forcing (ours) | 110.4 | 12.0 |

**Training and rollout cost comparison with baseline methods.** As shown in Table 7, we note our method is moderately slower than other SFT-based baselines but still within an acceptable range. This is mainly because GRPO-based fine-tuning requires the MLLM to generate multiple rollouts for each sample, whereas other methods only perform standard fine-tuning. In terms of rollout efficiency, our method is 1.6× faster than the SFT-based ZipVL and remains comparable to other methods, since MOBA and sharpness loss use manually fixed token budgets. However, as shown in Table 3, our method achieves a superior performance–efficiency tradeoff compared with SFT-based counterparts on both image and video benchmarks, making the additional training cost worthwhile.

Table 8: Additional comparison with temperature adjustment to enhance token sparsity. † denotes post-training MLLMs with ZipVL.

| Method | Ratio | MME | MMStar | ChartQA | VideoMME | Avg |
|---|---|---|---|---|---|---|
| Full | 100 | 2303 | 62.2 | 84.0 | 64.5 | 70.2 |
| MOBA | 25.0 | 1906 | 58.6 | 77.3 | 62.6 | 66.2 |
| ZipVL† | 61.7 | 2264 | 62.0 | 78.9 | 64.2 | 68.4 |
| Sharpness loss | 25.0 | 1965 | 59.6 | 77.0 | 63.7 | 66.8 |
| Temperature adjustment | 25.0 | 1960 | 59.8 | 78.6 | 61.3 | 67.4 |
| Sparsity Forcing(ours) | 26.4 | 2286 | 62.5 | 83.1 | 64.0 | 70.0 |

**Additional comparison with temperature adjustment baseline method.** Intuitively, temperature adjustment in the Softmax function can also sharpen the attention distribution. As shown in Table 8, temperature-based sharpening performs better than the sharpness-loss baseline, yet it still underperforms our method by 2.6%.

