# OpenReview forum: "Sparsity Forcing: Reinforcing Token Sparsity of MLLMs"
_ICLR.cc/2026/Conference — ICLR 2026 Poster_

### Official Review · Reviewer_DEv6 · 2025-10-31

**Soundness:** 3
**Presentation:** 2
**Contribution:** 3
**Rating:** 8
**Confidence:** 4

**Summary:**

The authors propose a token-sparse attention for VL with RL to enhance sparsity. It can be combined with different sparsity methods.

**Strengths:**

1. Sparsity is combined with RL, which is a new direction of sparsity training.
2. Evaluation of methods are solid, with different sparsity methods and models.

**Weaknesses:**

Personally I do not believe in token-sparsity with pruned KV cache as you never know what information is going to be used in the future. The situation is more severe when you have interaction with human, e.g. multi-round questioning/chating. Once you prune your KV cache, it's going to be lossy. As a result, it would be better either to use sparse attention kernel with full kv cache or you have certain ability to retrieve KV cache.

**Questions:**

1. What's the motivation behind "sharpness-inducing regularization/loss"? It makes no sense to me as you can tweek temperature (softmax scale) to adjust shapeness. The reason we use softmax scale in attention is that we want it to be "less sharp" otherwise zero attention score will lead to no grad for certain tokens and you miss the chance to correct your attention distribution when the model makes mistakes. Besides, in LLM, people use all kinds of tricks like QK norm or QK clip to reduce max-attn-logits (prevent too sharp) for training stablity.
2. How is moba baseline being trained? I believe moba is a pre-training method. If you apply its attn prob on a dense pretrain model, it should have large accuracy degradation.
3. Have you tried any non-pruned kv dynamic sparse method? For example. using SeerAttention like methods in post-training setting.
4. What's the cost of RL training-speed slowdown and GPU memory increase in current design? Similar to previous discussion, if you do not prune KV, your RL should also be efficient with a shared copy of KV cache. You only need to have differnt sparse index selections on different rollouts. Correct me if I am wrong.

---

> ### Author Response · Authors · 2025-11-24
> **Reply to Reviewer DEv6**
>
> Thanks to the reviewer for the valuable comments and insightful suggestions.
>
> **Q1: Personally I do not believe in token sparsity with a pruned KV cache, as you never know what information is going to be needed in the future. The situation is more severe when interacting with humans (e.g., multi-round questioning or chatting). Once you prune your KV cache, the lost information cannot be recovered. Therefore, it might be better to either use a sparse attention kernel with a full KV cache, or have some mechanism to retrieve KV cache when needed.**
>
> **A1:** We thank the reviewer for the insightful suggestion. We fully agree that in multi-round dialogue or long-horizon reasoning, aggressively pruning the KV cache during prefill can be lossy, as information deemed low-salience early on may later become necessary but cannot be recovered once removed.
>
> Following this suggestion, we have updated our implementation to adopt a retrieval-based KV-cache mechanism that is fully compatible with PagedAttention. This design retains the full KV cache and applies sparsity only through indexed retrieval, which not only substantially reduces training cost but also enables support for multi-round scenarios. Since this implementation parallelizes rollout generation without altering the optimization dynamics, its performance remains comparable to that of the previous implementation.
> Applying our method to the multi-turn setting is a promising direction, especially for VLA tasks that require long-term contextual retention, and we plan to further explore it in future work.
>
> **Q2: What's the motivation behind “sharpness-inducing regularization/loss”? It makes no sense to me as you can tweak temperature (softmax scale) to adjust sharpness. The reason we use a softmax scale in attention is that we want it to be *less* sharp; otherwise, zero attention scores will lead to no gradients for certain tokens, and you lose the chance to correct the attention distribution when the model makes mistakes. Besides, in LLMs, people use tricks like QK norm or QK clip to reduce max attention logits (i.e., prevent overly sharp attention) for training stability.**
>
>
> **A2:** We include the sharpness loss as a baseline to fairly represent a class of methods that attempt to “encourage sparsity via sharper attention’’ in the post-training regime, following prior work on entropy-/norm-based attention regularization. Intuitively, temperature adjustment in the Softmax function can also sharpen the attention distribution. As shown in the table below, temperature-based sharpening performs better than the sharpness-loss baseline, yet it still underperforms our method by 2.6%. The corresponding additional results can be found in Table 9 and are further analyzed in Section A.5 of our appendix.
>
> | Method                 | Ratio | MME  | MMStar | ChartQA | VideoMME | Avg  |
> |------------------------|-------|------|--------|---------|----------|------|
> | Full                   | 100   | 2303 | 62.2   | 84.0    | 64.5     | 70.2 |
> | MOBA                   | 25.0  | 1906 | 58.6   | 77.3    | 62.6     | 66.2 |
> | ZipVL                  | 61.7  | 2264 | 62.0   | 78.9    | 64.2     | 68.4 |
> | Sharpness loss         | 25.0  | 1965 | 59.6   | 77.0    | 63.7     | 66.8 |
> | Temperature adjustment | 25.0  | 1960 | 59.8   | 78.6    | 61.3     | 67.4 |
> | Sparsity Forcing       | 26.4  | 2286 | 62.5   | 83.1    | 64.0     | 70.0 |
>
>
> **Q3: How is the MOBA baseline being trained? I believe MOBA is a pre-training method. If you apply its attention probabilities on a dense pre-trained model, it should cause large accuracy degradation.**
>
> **A3:** Thank you for the question. Ideally, MOBA should indeed be applied during pre-training. However, we do not have access to the full pre-training pipeline of Qwen/Qwen-VL, so it is infeasible for us to re-pre-train the model from scratch with MOBA. Instead, we follow the original MOBA implementation as closely as possible and apply it in the post-training stage on Qwen2.5VL, using the same dataset as the other baselines.
>
> As you anticipated, this post-hoc application of MOBA on a dense pre-trained model does lead to noticeable performance degradation. In our experiments, the average score drops by about 4% compared to the original model (see Table 3), which further highlights the advantage of our RL-based method in preserving accuracy under sparse attention.

---

> > ### Author Response · Authors · 2025-11-25
> >
> > **Q4: Have you tried any non-pruned KV dynamic sparse method? For example, using SeerAttention-like methods in a post-training setting.**
> >
> > **A4:** Thank you for your suggestion. We further incorporate SeerAttention into our framework. Specifically, we first train the AttenGate module of SeerAttention within Qwen2.5VL, and then perform RL-based post-training with the frozen AttenGate and a trainable backbone model using our method. Both stages are trained on Video-R1-260k.
> >
> > During RL-based post-training, we adopt a Top-k strategy based on the gating scores to determine block-level attention sparsity. The rollout budgets are sampled from the range [25%, 50%] with a step size of 5%. During inference, we use a 25% budget.
> >
> > | Method               | Ratio | MME  | MMStar | ChartQA | VideoMME | Avg. |
> > |----------------------|-------|------|--------|---------|----------|------|
> > | SeerAttention + ours | 25%   | 2209 | 59.4   | 81.6    | 60.5     | 67.1 |
> > | ZipVL + ours         | 26.4  | 2286 | 62.5   | 83.1    | 64.0     | 70.0 |
> >
> >
> > As shown in the table above, ZipVL appears to be more compatible with our framework. We attribute this to the rigidity of the Top-k strategy used in online sparsity enforcement: because Top-k always preserves a fixed number of tokens per block, it provides less flexibility for probing low-salience tokens compared to the adaptive Top-p mechanism. This makes it harder for the RL policy to accurately evaluate which tokens can be safely pruned, leading to weaker overall performance.
> >
> >
> > **Q5: What's the cost of RL training-speed slowdown and GPU memory increase in the current design? Similar to the previous discussion, if you do not prune KV, your RL should also be efficient with a shared copy of the KV cache. You only need to have different sparse index selections for different rollouts.**
> >
> >
> > **A5:** We appreciate the suggestion and have updated our implementation accordingly. The main training-speed bottleneck in our previous design came from pruning the KV cache during RL rollouts. Because the KV cache was dynamically pruned, each rollout had to be decoded sequentially with its own KV cache, resulting in one-rollout-at-a-time generation and significantly slower training.
> >
> > As suggested, we now retain the full KV cache and apply sparsity only through different index selections during decoding via PagedAttention, allowing all rollouts to share the same cache and be generated in parallel. This eliminates the need to rebuild the KV cache for every rollout, greatly reducing training latency and GPU memory usage without affecting the optimization dynamics. In practice, this non-pruned KV design accelerates training by 6.3× and remains compatible with single-turn, multi-turn, and reasoning-intensive settings. We have updated the corresponding content in the Limitations and Future Work section accordingly.

---

> > > ### Comment · Reviewer_DEv6 · 2025-11-26
> > >
> > > Thanks for the insightful discussion. I have no more quesiton and will maintain my score.

---

### Official Review · Reviewer_92KG · 2025-11-01

**Soundness:** 3
**Presentation:** 3
**Contribution:** 2
**Rating:** 4
**Confidence:** 3

**Summary:**

The paper proposes Sparsity Forcing, a post-training, RL-based framework that explicitly trades off answer correctness and token savings for MLLMs. The method performs grouped, multi-budget rollouts using top-p sparse attention; a joint reward promotes correctness and token reduction, and GRPO (group-relative PPO) updates a sparse policy model while anchoring to a reference (full-attention) model for stability. Experiments on 13 image/video benchmarks claim to raise token reduction from ~20% to ~75% with minimal accuracy loss, yielding up to 3.0× memoryand 3.3× decoding speedups in long-context settings.

**Strengths:**

1. The proposed effieicency-aware RL post-training is important and novel.
2. The experimental results show the proposed method is very promising.
3. The paper writing is very clear and easy to follow.

**Weaknesses:**

1. The method needs to compute a_sort, nnz, topk_index. Some of these functions have linear complexity. It is hard to compute in parallel. How did the authors compute this function efficiently on GPU in Algorithm 1&2 ?
2. MOBA keeps 25% of full attention. Sparsity Forcing keeps 26.4%. It seems dynamic sparsity cannot achieve lower compression rate. Can the proposed method be applied to block-based patterns to achieve the best of both worlds?
3. How did the authors deal with the irregularity introduced by the dynamic sparsity? It is not very friendly to system deployment, especially considering the skewness of KVCache, tensor core optimization and continuous batching.

**Questions:**

Please see weaknesses.

---

> ### Author Response · Authors · 2025-11-21
> **Reply to Reviewer 92KG**
>
> Thanks to the reviewer for the valuable comments.
>
> **Q1: The method needs to compute a_sort, nnz, topk_index. Some of these functions have linear complexity. It is hard to compute in parallel. How did the authors compute this function efficiently on GPU in Algorithm 1&2 ?**
>
> **A1:** Thank you for the insightful comment. Although the operations used in our
> token-selection module (i.e., sorting a_sort, counting
> nnz(A[:,j]), and selecting tokens via topk_index) appear linear or difficult
> to parallelize in theory, in practice all of them are implemented using highly optimized GPU
> primitives. As a result, their runtime overhead is negligible compared with the cost of
> multi-head attention. See below for details:
>
> **1. Sorting accumulated attention scores.**
> We use GPU-parallel sorting (CUDA Thrust/CUB radix sort, as invoked by `torch.sort`). Since
> the number of tokens per layer is relatively small (typically a few thousand), this operation
> completes within microseconds and is not a performance bottleneck.
>
> **2. Counting non-zero entries (nnz(A[:,j]).**
> Because the attention matrix A is strictly lower-triangular under the causal mask, its
> pattern of non-zero entries is fully determined by the token index and sequence length, not
> by actual attention values. Thus, nnz(A[:,j] can be computed in O(1) time
> without explicitly calling `count_nonzero`. In practice, we precompute this vector once on
> GPU using a single  `torch.flip(torch.arange(1, Token_length+1, device=A.device))`   call and reuse it throughout training, incurring negligible overhead.
>
> **3. Top-k selection.**
> The topk_index operator uses CUDA-optimized selection routines (e.g., `torch.topk`), which
> run in parallel over the score vector. The GPU complexity is O(l), and its latency is
> insignificant compared with the O(l^2) FLOPs of attention.
>
> We also provide the decomposed TTFT results under different input lengths in the table below, showing that the cost of these operations is **negligible** relative to the overall speedup achieved by sparse attention.
>
> | Input Length | Method                     | TTFT (s) |
> |--------------|-----------------------------|----------|
> | **16K**      | Original                    | 3.50     |
> |              | + approximate attention     | 4.40     |
> |              | + sort & cumsum             | 4.42     |
> |              | + normalize & top-k         | 4.46     |
> |              | + sparse attention          | **2.85** |
> | **32K**      | Original                    | 9.41     |
> |              | + approximate attention     | 11.02    |
> |              | + sort & cumsum             | 11.03    |
> |              | + normalize & top-k         | 11.08    |
> |              | + sparse attention          | **5.92** |
>
>
> **Q2: MOBA keeps 25% of full attention. Sparsity Forcing keeps 26.4%. It seems dynamic sparsity
> cannot achieve lower compression rate. Can the proposed method be applied to block-based
> patterns to achieve the best of both worlds?**
>
> **A2:** We thank the reviewer for this insightful comment. The 25% retention ratio used for MOBA is a
> *manually fixed* hyperparameter designed to place MOBA in a comparable “low-budget”
> operating regime. In contrast, the 26.4% retention ratio observed in our method is *not*
> manually chosen; it is determined by the reward-driven policy that dynamically discovers the
> minimal token set required to preserve correctness. We can also adjust the hyperparameter
> p to achieve a lower budget, as shown in Figure 4(a).
>
> Moreover, as shown in Table 3, although MOBA retains slightly fewer tokens (25% vs. 26.4%),
> its accuracy decreases by 4%, while Sparsity Forcing remains comparable to full-attention
> performance. This indicates that our method achieves a better performance–efficiency
> trade-off.
>
> Regarding the suggestion about block-based patterns: our RL framework is orthogonal to the
> underlying sparse attention mechanism and can indeed operate on block-structured sparsity.
> Sparsity Forcing only determines *which* tokens should be kept; the specific sparsity kernel
> (token-level or block-level) can be interchanged without modifying the algorithm. Moreover,
> MOBA’s block-wise probing strategy provides an efficient approximation of attention and is
> compatible with our method. In future work, we plan to incorporate such block-wise probing
> to further reduce overhead and enable tighter integration with structured sparsity patterns.

---

> > ### Author Response · Authors · 2025-11-21
> >
> > **Q3: How to deal with the irregularity introduced by the dynamic sparsity? It is not very
> > friendly to system deployment, especially considering the skewness of KVCache, tensor core
> > optimization and continuous batching.**
> >
> > **A3:** We appreciate the reviewer's concern regarding the system-level implications of dynamic,
> > input-dependent sparsity. Our method is designed with practical deployment constraints in
> > mind, and in fact aligns naturally with existing high-throughput serving infrastructures.
> >
> > **First**, our approach operates at the token-level granularity, which is fully compatible with
> > *PagedAttention*. Therefore, Sparsity Forcing can be seamlessly integrated into widely
> > adopted serving systems such as *vLLM* and *SGLang* without requiring architectural
> > modification. Since these systems already rely on indirection-based KV cache management,
> > the irregularity introduced by selecting a variable number of active tokens per request is
> > handled naturally by their paged memory abstraction.
> >
> > **Second**, our sparse computation pattern does not interfere with common system
> > optimizations. Techniques such as prefix sharing, multi-phase attention, and continuous
> > batching remain compatible, because our sparse attention mechanism simply selects a subset
> > of tokens to participate in attention, without altering kernel interfaces or execution
> > schedules. This yields a flexible and efficient computation flow while preserving the
> > system-level invariants required for high throughput.
> >
> > **Finally**, the potential KV-cache skewness is mitigated by our page-based pruning
> > strategy: pruned tokens simply do not allocate new KV pages, while retained tokens continue
> > to use contiguous pages managed by the underlying runtime. Thus, the dynamic sparsity does
> > not introduce fragmentation beyond what PagedAttention is already designed to handle.

---

> ### Author Response · Authors · 2025-11-28
>
> Dear Reviewer 92KG
>
> Thank you again for your valuable comments regarding the deployment aspects of our method. In the revised manuscript and rebuttal, we have provided a detailed latency analysis and implementation details to address these concerns. We hope this clarifies the practical feasibility of our approach. Please let us know if you have any further questions or suggestions.
>
> Best regards,
>
> Authors

---

### Official Review · Reviewer_JKcD · 2025-11-01

**Soundness:** 2
**Presentation:** 2
**Contribution:** 2
**Rating:** 4
**Confidence:** 3

**Summary:**

This paper proposes Sparsity Forcing, an RL-based post-training framework that promotes token sparsity in multimodal LLMs using GRPO. The method treats token reduction and correctness as joint rewards. Applied on QwenVL and LLaVA-Video, it improves sparsity with minor accuracy loss.

**Strengths:**

Demonstrates sparsity gains across many MLLM tasks.

Clear implementation using grouped rollouts and ZipVL backbone.

Experiments across benchmarks covering both image and video. There are some interesting discussions in ablation study.

**Weaknesses:**

However, while the results are promising, I find the methodological novelty to be somewhat limited. The work mainly applies the existing GRPO framework to a known sparse attention mechanism (ZipVL/MOBA style) and does not clearly articulate what new algorithmic insights it introduces beyond this combination.

The paper also does not sufficiently discuss how sparsity behaviors differ between text-only LLMs and multimodal models, even though modality-specific sparsity patterns and cross-frame visual dependencies are central challenges in MLLMs.

The comparison to baselines leaves unanswered questions. Although the related work section lists many sparse attention approaches, only a small subset is used in experiments. It is not clear whether the baselines are retrained or used as training-free methods, nor whether the training and rollout costs are comparable. Since the method uses RL and multiple rollouts with a reference model, training cost is a key factor, and it would be helpful to discuss fairness and overhead.

In addition, the improvements over ZipVL are sometimes modest (Fig 4 cd), particularly in latency, which suggests that the marginal efficiency gain relative to the additional training cost may be less significant in some settings.

There are also concerns regarding generality and deployment. It appears that the method largely supports single-turn inference and requires recomputing context when multiple iterations occur, since the KV cache pruning changes across runs. This limits applicability to interactive or long-horizon reasoning settings.

The method also seems to require training for each sparsity level, which affects flexibility in real-world deployment scenarios where dynamic sparsity adjustment may be desired.

For video tasks, the approach applies the same sparsity policy as images and does not attempt to leverage temporal redundancy or cross-frame consistency, which misses an opportunity.

Finally, although the paper reports latency and memory results, the reward signal itself uses token ratio rather than actual measured runtime efficiency. This gap raises questions regarding hardware alignment, especially given that sparse attention can have non-linear speed behavior depending on the implementation.

**Questions:**

Do baseline methods also involve training? If so, what is the training and rollout cost for each? If not, how do you ensure comparison fairness?

Does this approach support multi-turn conversations with incremental KV cache usage, or does each turn require full recomputation?

Is the model retrained for each desired sparsity level, or can sparsity be adjusted at inference time without retraining?

How does the method handle video temporal redundancy, and could temporal attention provide additional efficiency gains?

Why not include real hardware latency as part of the efficiency reward instead of only token reduction?

---

> ### Author Response · Authors · 2025-11-21
> **Reply to Reviewer JKcD**
>
> Thanks to the reviewer for the valuable comments.
>
> **Q1: While the results are promising, I find the methodological novelty to be somewhat limited. The work mainly applies the existing GRPO framework to a known sparse attention mechanism (ZipVL/MOBA style) and does not clearly articulate what new algorithmic insights it introduces beyond this combination.**
>
> **A1:** While our framework builds on GRPO and adopts ZipVL as a sparsity-friendly attention
> backbone, the key novelty of our work lies in a fundamentally different way of *optimizing*
> token sparsity. The specific technical contributions that enable this “forced sparsity”
> paradigm are as follows:
>
> 1. **Progressive budget sweep (multi-budget rollouts).**
>    Standard GRPO samples diverse answers randomly. In contrast, we introduce a structured
>    multi-budget rollout mechanism tailored for the efficiency–accuracy tradeoff. For each
>    input, we execute a group of rollouts under different sparsity thresholds \(p\). By
>    contrasting their correctness within the group, the reward identifies the *tipping point*
>    at which additional pruning begins to harm accuracy, providing a direct, data-driven
>    signal for which tokens are essential.
>
> 2. **Overcoming the “proxy objective” problem in sparse training.**
>    Prior sparsity methods rely on proxy losses—e.g., attention sharpness, entropy
>    minimization, or \(L_{\infty}\) regularization—to encourage sparsity. However, such proxies
>    do not reliably correlate with answer correctness or actual compute savings and often
>    impose rigid patterns independent of layer-wise dynamics. As shown in Table 3, these
>    methods achieve a poorer performance–efficiency tradeoff than ours on both image and
>    video benchmarks. We instead formulate efficiency (token reduction) and accuracy (reward
>    correctness) as a joint end-to-end RL objective. No differentiable sparsity proxy is
>    required: the model is penalized only when unnecessary tokens lead to computational waste
>    or an incorrect answer.
>
> 3. **Eliminating the training–inference mismatch via RL.**
>    Existing trainable sparse-attention methods (e.g., MOBA, NSA) rely on supervised
>    fine-tuning with teacher forcing, causing the model to experience error accumulation when
>    information is missing. Our Sparsity Forcing framework is inference-aligned: RL rollouts
>    are generated autoregressively *with the pruning policy active*, allowing the model to
>    explicitly learn to answer correctly under its own pruned KV-cache. This results in
>    deployment-ready sparsity without the robustness drops observed in SFT baselines (as
>    shown in Table 3).
>
> In summary, our novelty is not the invention of GRPO or ZipVL, but the proposed *Sparsity
> Forcing* framework that repurposes these tools to address the core failure modes of passive
> pruning via attention sharpness and SFT-based pruning, yielding a principled RL-driven
> solution to the accuracy–efficiency tradeoff.

---

> ### Author Response · Authors · 2025-11-21
>
> **Q2: Do baseline methods also involve training? If so, what is the training and rollout cost for each? If not, how do you ensure comparison fairness?**
>
> **A2:** Thank you for the question. All baseline methods we compare against are training-based approaches. The detailed implementation for the sharpness-based loss is provided in Section A.2 of our appendix for completeness. For MOBA and ZipVL, we follow their respective papers and post-train
> Qwen2.5VL accordingly.
>
> To ensure fairness, we train all baseline methods on the same datasets as our method. We
> also report the wall-clock training time and latency under a 128k context for each method,
> as summarized in the table below. Our method incurs a moderately higher training cost
> because RL-based fine-tuning requires multiple rollouts per sample, whereas the other
> baselines rely solely on standard supervised fine-tuning. In terms of rollout efficiency, our
> method is 1.6× faster than the SFT-based ZipVL and remains comparable to other methods,
> since MOBA and sharpness loss use manually fixed token budgets.
>
> Nevertheless, our approach achieves a substantially better performance–efficiency tradeoff than all baseline methods, making the additional training cost well justified. The corresponding additional results can be found in Table 8 and Section A.5 of our appendix.
>
> | Method                  | Ratio | MME  | MMStar | ChartQA | VideoMME | Avg  | training time (hour) | Latency (s) |
> |-------------------------|-------|------|--------|---------|----------|------|----------------------------------|-------------|
> | Full                    | 100   | 2303 | 62.2   | 84.0    | 64.5     | 73.2 | --                               | --          |
> | MOBA                    | 25.0  | 1906 | 58.6   | 77.3    | 62.6     | 66.6 | 75.6                            | 11.6        |
> | ZipVL                   | 61.7  | 2264 | 62.0   | 78.9    | 64.2     | 71.5 | 81.0                            | 18.7        |
> | Sharpness loss          | 25.0  | 1965 | 59.6   | 77.0    | 63.7     | 67.6 | 88.4                            | 11.9        |
> | Sparsity Forcing (ours) | 26.4  | 2286 | 62.5   | 83.1    | 64.0     | 72.8 | 110.4                           | 12.0        |
>
>
> **Q3: Does this approach support multi-turn conversations with incremental KV cache usage, or does each turn require full recomputation?**
>
> **A3:**  Thank you for the comment. Our method naturally supports multi-turn
> conversations with incremental KV cache usage. From the system perspective, each new turn
> does *not* require recomputing all previous context. The KV cache from earlier turns is
> stored and can be retrieved directly. During decoding, we apply sparsity only to the cached
> keys/values through a probing strategy combined with paged-attention–style indexing. This
> allows us to selectively fetch the relevant KV from previous turns without rebuilding the
> entire cache.
>
> **Q4: Is the model retrained for each desired sparsity level, or can sparsity be adjusted at inference time without retraining?**
>
> **A4:** Thank you for the comment. Our method does *not* require retraining the model for each desired sparsity level. During RL training, different rollouts use different sparsity levels, enabling the model to actively explore the optimal sparsity level that preserves accuracy. At inference time, the sparsity level can be flexibly adjusted by the threshold parameter p. As shown in Fig. 4(a), varying p allows us to smoothly control the sparsity ratio without any additional training.
>
> **Q5: How does the method handle video temporal redundancy, and could temporal attention provide additional efficiency gains?**
>
> **A5:**  Thank you for the suggestion. Our current method addresses temporal redundancy implicitly through
> the attention-based token selection mechanism. Since we select tokens based on accumulated
> attention scores over the sequence, these scores already encode temporal relationships
> across frames: temporally redundant tokens tend to receive consistently lower attention and
> are therefore more likely to be skipped, while tokens on key frames with important motion or
> scene changes are preserved. Empirically, this behavior is reflected in Fig. 4(b), where
> increasing the number of frames (i.e., the sequence length) allows our method to operate under
> a lower token budget while maintaining similar accuracy.

---

> > ### Author Response · Authors · 2025-11-26
> >
> > **Q6: Why not include real hardware latency as part of the efficiency reward instead of only token reduction?**
> >
> > **A6:** Thank you for the suggestion. We agree that real hardware latency is an important practical
> > metric. Our main reason for using the token reduction ratio as the efficiency reward is that it
> > is hardware-agnostic and tightly coupled with FLOPs and KV-cache usage, whereas real latency
> > can be noisy and system-dependent.
> >
> > To verify this design choice, we additionally conducted an ablation where we include hardware
> > latency in the efficiency reward. Concretely, we compare two variants on LLaVA-Video-7b: (i) using only the token reduction ratio as the efficiency term, and  (ii) using a combined reward of latency and token reduction ratio.
> >
> > | Efficiency Reward               | Ratio | VideoMME    | MLVU | VideoMMMU | PerceptionTest | EgoSchema | TempCompass | Avg. |
> > |--------------------------------|-------|-------------|-----------------------|------------|-----------------|-----------|-------------|------|
> > | Token Reduction Ratio          | 29.6  | 64.1/70.2   | 70.3                  | 35.2       | 66.0            | 57.0      | 66.2        | 61.3 |
> > | Latency + Token Reduction Ratio | 30.5  | 63.9/70.3   | 70.4                  | 35.0       | 66.4            | 56.4      | 66.0        | 61.2 |
> >
> > As shown in the table above, incorporating latency into the reward does not bring noticeable
> > benefits: both variants achieve almost identical accuracy (61.3 vs. 61.2 Avg.) with very
> > similar token ratios. These findings suggest that the token reduction ratio is a sufficiently
> > good surrogate for hardware efficiency in our setting. The corresponding additional results
> > can be found in Table 7 and are further analyzed in Section A.5 of our appendix.

---

> > > ### Author Response · Authors · 2025-11-28
> > >
> > > Dear Reviewer JKcD:
> > >
> > > Your feedback has been invaluable in helping us refine and strengthen the manuscript. We have incorporated all suggestions and made substantial improvements throughout. We would be grateful if you would consider updating your evaluation to reflect these revisions. Thanks for your time, thoughtful comments, and contribution to improving this work.
> > >
> > > Best regards,
> > >
> > > Authors

---

### Official Review · Reviewer_htVe · 2025-11-03

**Soundness:** 3
**Presentation:** 3
**Contribution:** 2
**Rating:** 4
**Confidence:** 3

**Summary:**

The paper identifies a real and important limitation of current sparse-token methods for MLLMs and they rely on emergent sparsity and plateau around ~50% token reduction.

**Strengths:**

1. Transforms token sparsity into an explicit joint reward (accuracy + sparsity) and performs GRPO over multiple budget rollouts, effectively avoiding the token-sparsity mismatch seen in SFT-based methods.

2. General and practical framework that requires no architectural modifications and remains fully compatible with ZipVL and other sparse-attention approaches.

**Weaknesses:**

1. Is the reward shaping universally effective, or does it depend on dataset and model scale? Additional experiments or analysis on scalability and robustness would strengthen the claims.

2. Although the method’s goal is to improve inference efficiency, the paper does not disclose critical training cost details. Specifically:

    2.1 Total training time?

    2.2 GPU resources (e.g., A100 hours)?

    2.3 Wall-clock comparison vs. ZipVL, LoRA, or R1-style RL fine-tuning?

**Questions:**

see above.

---

> ### Author Response · Authors · 2025-11-21
> **Reply to Reviewer htVe**
>
> Thanks to the reviewer for the valuable comments.
>
>
> **Q1: Is the reward shaping universally effective, or does it depend on dataset and model scale? Additional experiments or analysis on scalability and robustness would strengthen the claims.**
>
> **A1:** Thank you for your review. Our Sparsity Forcing method is a general
> sparsity-enhancing framework and is not tailored to any specific model size or dataset.
> As shown in Table 1 and 2, it generalizes well across different model scales (3B and 7B). We further
> apply our method to LLaVA-OV [1] to evaluate the scalability of the reward design across
> different model sizes and datasets, demonstrating its broad applicability.
>
> | Model            | Method | Calibration Dataset     | Ratio  | MME  | MMStar | ChartQA | MMBench | VideoMME   | EgoSchema | Avg  |
> |------------------|--------|--------------------------|--------|------|--------|---------|---------|-------------|-----------|------|
> | Qwen2.5VL-3B     | Full   | –                        | 100%   | 2157 | 55.5   | 83.4    | 78.8    | 61.2/66.8   | 64.2      | 69.6 |
> | Qwen2.5VL-3B     | Ours   | Video-R1-260k            | 24.1%  | 2146 | 55.5   | 83.2    | 78.4    | 61.0/66.5   | 63.7      | 69.3 |
> | Qwen2.5VL-7B     | Full   | –                        | 100%   | 2303 | 62.2   | 84.0    | 83.9    | 64.5/71.1   | 64.3      | 73.2 |
> | Qwen2.5VL-7B     | Ours   | Video-R1-260k            | 25.3%  | 2286 | 62.5   | 83.1    | 84.1    | 64.0/71.0   | 64.2      | 72.9 |
> | LLaVA-OV-0.5B    | Full   | –                        | 100%   | 1476 | 37.5   | 61.2    | 52.1    | 44.0/43.5   | 26.6      | 45.4 |
> | LLaVA-OV-0.5B    | Ours   | LLaVA-OneVision-Data     | 23.5%  | 1463 | 37.1   | 60.5    | 52.0    | 43.8/43.2   | 26.5      | 45.1 |
> | LLaVA-OV-7B      | Full   | –                        | 100%   | 1998 | 61.5   | 80.0    | 80.5    | 58.0/61.4   | 60.0      | 67.5 |
> | LLaVA-OV-7B      | Ours   | LLaVA-OneVision-Data     | 26.7%  | 1994 | 61.4   | 79.2    | 80.6    | 57.9/61.6   | 59.4      | 67.3 |
>
> As shown in the above table, our method consistently preserves performance while reinforcing token sparsity across different model sizes (0.5B, 3B, and 7B). Exploring larger model scales is left for future work. In addition, our method works well across different calibration datasets, such as Video-R1-260k and LLaVA-OneVision-Data, demonstrating its scalability with respect to data diversity. The corresponding additional results can be found in Table 6 and are further analyzed in Section A.5 of our appendix.
>
> [1] Li B, Zhang Y, Guo D, et al. Llava-onevision: Easy visual task transfer[J]. arXiv preprint arXiv:2408.03326, 2024.
>
>
> **Q2: Although the method’s goal is to improve inference efficiency, the paper does not disclose critical training cost details. Specifically: (1) Total training time (2) GPU resources (e.g., A100 hours) (3) Wall-clock comparison vs. ZipVL, MOBA, or R1-style RL fine-tuning.**
>
> **A2:** Our method is trained on 8 A100 GPUs with 883 GPU hours for Qwen2.5VL-7b and 164 GPU hours for LLaVA-Video-7b. We also compare the training wall-clock time with other baseline methods.
>
> | Method                   | Ratio |  MME | MMStar | ChartQA | VideoMME |  Avg | Wall-clock time for training (hours) |
> |--------------------------|:-----:|-----:|:------:|:-------:|:--------:|-----:|--------------------------------------:|
> | Full                     | 100.0 | 2303 |  62.2  |  84.0   |   64.5   | 73.2 | N/A                                  |
> | MOBA                     |  25.0 | 1906 |  58.6  |  77.3   |   62.6   | 66.6 | 75.6                                 |
> | ZipVL                    |  61.7 | 2264 |  62.0  |  78.9   |   64.2   | 71.5 | 81.0                                 |
> | Sharpness loss           |  25.0 | 1965 |  59.6  |  77.0   |   63.7   | 67.6 | 88.4                                 |
> | Sparsity Forcing (ours)  |  26.4 | 2286 |  62.5  |  83.1   |   64.0   | 72.8 | 110.4                                |
>
>
> As shown in the above table, our method is moderately slower than other SFT-based baselines but still within an acceptable range. This is mainly because R1-style fine-tuning requires the MLLM to generate multiple rollouts for each sample. However, our method achieves a superior performance–efficiency tradeoff compared with SFT-based counterparts on both image and video benchmarks, making the additional training cost worthwhile. For example, our method outperforms MOBA with 4% on average while further reducing 35.3% tokens with respect to ZipVL. The corresponding additional results can be found in Table 8 and are further analyzed in Section A.5 of our appendix.

---

> > ### Comment · Reviewer_htVe · 2025-11-25
> >
> > Thanks for your reply. I will keep my score because of the inefficiency compared with other SOTA methods.

---

> > > ### Author Response · Authors · 2025-11-25
> > >
> > > We believe there is a misunderstanding of our results, and we clarify them here. Our approach achieves performance comparable to the original model on 13 benchmarks while using only 25% of the inference budget, yielding a substantially better performance–efficiency trade-off than both training-free and training-based methods.
> > >
> > > We respectfully emphasize that the ultimate goal of our work is to secure the best possible inference efficiency while maintaining high performance. The training overhead from RL is a one-time investment that yields the superior performance-efficiency tradeoff during deployment. Practically, the training cost (110 hours) remains fully acceptable (completable within a reasonable timeframe on a standard 8-GPU node, similar to SFT baselines (88 hours)), which we believe is justified by the significant, recurring gains in inference speed and memory reduction.

---

> > > > ### Author Response · Authors · 2025-11-28
> > > >
> > > > Dear Reviewer htVe:
> > > >
> > > > Thank you again for your thoughtful review and comments on the training efficiency of our method. We believe there may have been some misunderstanding, and we have clarified this point in our rebuttal. If you have any further questions or would like to discuss this aspect in more detail, please feel free to let us know.
> > > >
> > > > Best regards,
> > > >
> > > > Authors

---

### Meta-Review · Area_Chair_9hri · 2026-01-06

**Summary:**

Reviewers' concerns including limited novelty, comparison with baselines, generality and deployment, reward signal design and so on. Most are addressed by authors' rebuttal.
Reviewer htVe believes that the proposed method is inefficient compared with other SOTA methods since it requires more time for training. However, I agree with authors that the training overhead from RL is a one-time investment that yields the superior performance-efficiency tradeoff during deployment.

**Reviewer Concerns:**

The authors' rebuttal and extra experiments address most concerns including more base models and training datasets, reward signal design and comparison with other methods.
The proposed method is not superior to the baseline in all aspects. It has longer training time and slightly higher ratio (26.4 vs. 25.0). However, I believe that the proposed method is a practical approach and more effective than previous methods.
Also, although the authors re emphasizes the novelty, it is not sure whether reviewer JKcD is convinced since there is no participation. Personally, I think this article explores a new direction of sparsity training.

**Reviewer Scores:**

This article has original scores of 4,4,4,8. One reviewer maintained the score of 4. One reviewer maintained the score of 8, and the others didn't participate in the discussion.

---

### Decision · Program_Chairs · 2026-01-26

Accept (Poster)